# Cell type specificity of neurovascular coupling in cerebral cortex

Hana Uhlirova[1†‡], Kıvılcım Kılıç[2†], Peifang Tian[2,3], Martin Thunemann[1], Michèle Desjardins[1], Payam A Saisan[2], Sava Sakadžić[4], Torbjørn V Ness[5], Celine Mateo[6], Qun Cheng[2], Kimberly L Weldy[2], Florence Razoux[2], Matthieu Vandenberghe[1,7], Jonathan A Cremonesi[8], Christopher GL Ferri[2], Krystal Nizar[9], Vishnu B Sridhar[10], Tyler C Steed[9], Maxim Abashin[11], Yeshaiahu Fainman[11], Eliezer Masliah[2], Srdjan Djurovic[12,13], Ole A Andreassen[7], Gabriel A Silva[10,14], David A Boas[4], David Kleinfeld[6,11,15], Richard B Buxton[1], Gaute T Einevoll[5,16], Anders M Dale[1,2], Anna Devor[1,2,4*]

[1]Department of Radiology, University of California, San Diego, La Jolla, United States; [2]Department of Neurosciences, University of California, San Diego, La Jolla, United States; [3]Department of Physics, John Carroll University, University Heights, United States; [4]Martinos Center for Biomedical Imaging, Harvard Medical School, Charlestown, United States; [5]Department of Mathematical Sciences and Technology, Norwegian University of Life Sciences, Ås, Norway; [6]Department of Physics, University of California, San Diego, La Jolla, United States; [7]NORMENT, KG Jebsen Centre for Psychosis Research, Division of Mental Health and Addiction, University of Oslo, Oslo, Norway; [8]Biology Undergraduate Program, University of California, San Diego, La Jolla, United States; [9]Neurosciences Graduate Program, University of California, San Diego, La Jolla, United States; [10]Department of Bioengineering, University of California, San Diego, La Jolla, United States; [11]Department of Electrical and Computer Engineering, University of California, San Diego, La Jolla, United States; [12]Department of Medical Genetics, Oslo University Hospital, Oslo, Norway; [13]NORMENT, KG Jebsen Centre for Psychosis Research, Department of Clinical Science, University of Bergen, Bergen, Norway; [14]Department of Ophthalmology, University of California, San Diego, La Jolla, United States; [15]Section of Neurobiology, University of California, San Diego, La Jolla, United States; [16]Department of Physics, University of Oslo, Oslo, Norway

**\*For correspondence:** adevor@ucsd.edu

[†]These authors contributed equally to this work

**Present address:** [‡]Central European Institute of Technology, Faculty of Mechanical Engineering, Brno University of Technology and Institute of Physical Engineering, Brno, Czech Republic

**Abstract** Identification of the cellular players and molecular messengers that communicate neuronal activity to the vasculature driving cerebral hemodynamics is important for (1) the basic understanding of cerebrovascular regulation and (2) interpretation of functional Magnetic Resonance Imaging (fMRI) signals. Using a combination of optogenetic stimulation and 2-photon imaging in mice, we demonstrate that selective activation of cortical excitation and inhibition elicits distinct vascular responses and identify the vasoconstrictive mechanism as Neuropeptide Y (NPY) acting on Y1 receptors. The latter implies that task-related negative Blood Oxygenation Level Dependent (BOLD) fMRI signals in the cerebral cortex under normal physiological conditions may be mainly driven by the NPY-positive inhibitory neurons. Further, the NPY-Y1 pathway may offer a potential therapeutic target in cerebrovascular disease.

**eLife digest** Unlike other cells in the body, brain cells contain almost no energy reserves and rely on blood vessels for continuous supply of oxygen. A change in the brain's activity can cause these blood vessels to either dilate or constrict, which alters the supply to match the change in demand. However, it is not known which signals cause these changes in the blood vessels.

Previous studies have shown that individual blood vessels in an intact brain tend to dilate when the brain's activity increases, and constrict when brain activity is inhibited. However, these studies were based on correlations, and there was no direct evidence that the inhibitory cells cause blood vessels to constrict.

Uhlirova, Kılıç et al. now provide such evidence. The experiments made use of mice that had been genetically modified such that the excitatory or inhibitory nerve cells in their brains could be selectively activated by shining a blue light on the brain's surface. The vessels in the outer millimeter of the gray matter of each mouse's brain were imaged in detail, both before and after the blue light was used to activate the nerve cells.

The experiments reveal that both excitatory and inhibitory nerve cells can cause blood vessels in the brain to dilate. However, blood vessels in the brain will only constrict in response to inhibitory nerve cells. Uhlirova, Kılıç et al. went on to identify a molecule called Neuropeptide Y (or NPY short) as a signal that triggers the constriction of the blood vessels. This signaling molecule is released by a specific sub-type of inhibitory nerve cell and it binds to a receptor protein on the brain's blood vessels to make them constrict.

These findings suggest that NPY and its receptor on blood vessels may offer promising targets for drugs to treat diseases of the brain's blood vessels. Further studies are now needed to identify the signals responsible for the dilation of blood vessels in the brain.

## Introduction

In the past decade, the field of cerebral blood flow and metabolism has experienced a paradigm shift with respect to neurovascular coupling mechanisms. An earlier 'metabolic' hypothesis postulated that an increase in cerebral blood flow (CBF) in response to increased neuronal activity (a.k.a. 'functional hyperemia') was directly related to energetic costs and driven by bi-products of energy consumption ($CO_2$, lactate, $H^+$, etc.) (reviewed in [*Raichle and Mintun, 2006*]). However, a growing body of experimental evidence, including our own, indicates that while molecules produced by increased energy metabolism do have a vasoactive effect, much of the acute CBF response in vivo under healthy conditions is driven by vasoactive messengers related to neuronal signaling (for recent reviews see [*Kleinfeld et al., 2011*; *Cauli and Hamel, 2010*; *Buxton et al., 2014*]). These messengers, released by specific cell types, actively regulate arteriolar diameters – a key control parameter in the CBF response. Thus, activation of specific types of neurons (and potentially astrocytes) as opposed to the undifferentiated spiking or synaptic activity is likely to determine the vascular response.

Previous studies provided evidence for the involvement of both excitatory and inhibitory neurons in CBF regulation. In Pyramidal cells (PCs), activation of N-methyl-D-aspartate (NMDA) receptors has been demonstrated to stimulate release of prostaglandin E2 produced by cyclooxygenase-2 (COX-2) causing an increase in CBF (*Lacroix et al., 2015*; *Lecrux et al., 2011*). In inhibitory neurons (INs) – that can be further classified into subtypes based on their neurotransmitter/neuropeptide content, shape, and neurophysiological properties (*Markram et al., 2004*) – release of neuropeptides and nitric oxide (NO) has been hypothesized to provide a bidirectional CBF control (reviewed in [*Cauli and Hamel, 2010*]). Indeed, experiments in cortical brain slices have demonstrated that stimulation of INs can cause dilation or constriction of arteriolar segments embedded in the sliced tissue with the polarity of the effect depending on the INs cell type (*Cauli et al., 2004*; *Perrenoud et al., 2012*). Further, selective optogenetic (OG) stimulation of INs in vivo was shown to increase CBF (*Anenberg et al., 2015*).

On the level of single cortical arterioles, two-photon imaging studies have revealed that the hemodynamic response to a sensory stimulus is composed of a combination of dilatory and

constrictive phases with the relative strength of vasoconstriction co-varying with that of neuronal inhibition (*Devor et al., 2007*, *2008*; *Tian et al., 2010*). However, direct evidence that activation of INs can cause arteriolar vasoconstriction in vivo is missing. Therefore, in the present study, we asked whether selective OG stimulation of INs can cause the biphasic dilation/constriction sequence characteristic of sensory-induced responses and whether the constriction phase was specific to activation of INs rather than excitatory neurons. Our results confirm the ability of INs to drive the biphasic arteriolar response and provide the first in vivo evidence for the major role of the NPY-Y1 pathway mediating vasoconstriction.

## Results

### Sensory stimulation induces arteriolar dilation followed by constriction with the fastest dilation onset below layer IV

Previously, we and others have shown that sensory stimuli in the primary somatosensory cortex (SI) induced biphasic changes in arteriolar diameters – dilation followed by constriction – with the onset and time-to-peak of dilation depending on the cortical depth (*Tian et al., 2010*; *Lindvere et al., 2013*). In these studies, imaging was performed in the rat SI within the top ~500 µm (layers I-IV in the rat SI), and the fastest dilation occurred at the deepest measurement locations. Since

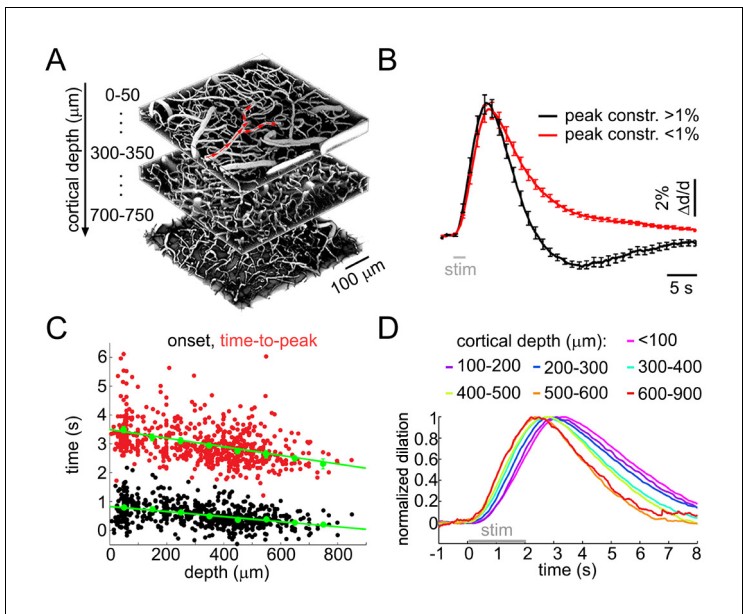

**Figure 1.** Sensory stimulus-induced arteriolar response. (**A**) An example vascular image stack throughout the cortical depth. Three 50-µm slabs at different depths are shown. Red arrows indicate the direction of flow in a surface arteriole diving at 3 points. (**B**) Sensory stimulus-induced dilation time-courses sorted into two categories by the presence or absence of the constriction phase, defined as the peak constriction amplitude exceeding 1%. The black curve (constriction): n = 419 measurements, 34 subjects. The red curve (no constriction): n = 397 measurements, 47 subjects. Error bars indicate standard error (SE) across subjects. (**C**) Onset (black) and time-to-peak (red) of dilation as a function of depth. Each data point represents a single measurement. For each subject, the data were group-averaged according to depth in 100-µm bins. Error bars represent the mean ± SE across subjects for each bin (green). (**D**) A zoomed-in view onto the initial eight seconds of the response. Time-courses were grouped by depth as in (**C**) and peak-normalized (see Materials and methods) to facilitate visual inspection of temporal differences. Color-coded depth categories are indicated on top.

The following figure supplement is available for figure 1:

**Figure supplement 1.** Sensory stimulus-induced arteriolar response – additional quantification.

infragranular layers V and VI were not sampled, it remains unclear whether sensory-induced dilation initiated in layer IV (which has the highest metabolic load [*Sokoloff et al., 1977*]) or below.

In the present study, we extended these measurements down to ~900 µm (layer VI) in the mouse SI (*Figure 1A*). This was possible due to utilization of longer illumination wavelengths (see Materials and methods) and higher transparency of the mouse cortex compared to the rat; this is due to smaller diameter of the surface vessels attenuating light. The stimulus consisted of a 2-s train of electrical pulses (100 µs, 1 mA, 3 Hz) delivered to the contralateral forepaw. Fluorescein isothiocyanate (FITC)-labeled dextran was injected intravenously to visualize the vasculature. Single-vessel diameter measurements were performed at 866 locations along 120 diving arterioles and their branches (n = 320 branch measurements) in 49 wild type subjects at depths from 20 to 850 µm. Consistent with our previous data, many of the measurements exhibited a biphasic arteriolar diameter change: dilation followed by constriction before returning to the baseline. To quantify the constriction amplitude, individual time-courses were temporally smoothed with a Gaussian kernel (FWHM = 0.5 s); the peak constriction was determined as the minimum value within 3–13 s after the stimulus onset. About half of the measurements showed constriction with the peak constriction amplitude exceeding 1% (*Figure 1B* and *Figure 1—figure supplement 1A*).

To quantify the dilation onset, we fit a straight line to the rising slope of the diameter increase between 20 and 80% of the peak and calculated an intercept with the pre-stimulus baseline for each measured time-course. There was a gradual decrease in the onset and time-to-peak throughout the depth (p=1.8 $\times$ 10$^{-6}$ and 2.3 $\times$ 10$^{-6}$, respectively; *Figure 1C–D*). The dilation amplitude varied between measurement locations (*Figure 1—figure supplement 1B,C*).

These results are in agreement with our earlier observations in the rat SI, demonstrating biphasic arteriolar diameter changes with depth-dependent dilation onset and peak. Novel to the present study, these data show that the fastest dilation onset and rise occurs in deep cortical layers, below layer IV.

## OG stimulation of INs reproduces the biphasic arteriolar response

We imaged OG-induced single-vessel diameter changes in 17 VGAT-ChR2(H134R)-EYFP mice (*Zhao et al., 2011*). Double immunofluorescence of ChR2-EYFP and GABA-producing glutamate decarboxylase in this mouse line revealed a virtually complete overlap of staining in the SI confirming that the ChR2-EYFP protein was indeed present only in GABAergic INs (*Figure 2—figure supplement 1*). The OG stimulus was delivered to the cortical surface through the objective using a cylinder-shaped 473-nm laser beam 230 µm in diameter (FWHM) (*Figure 2A*). Due to significant absorption of light at this (blue) wavelength, the power of the beam was reduced by ~ 90% within the first 200 µm, practically confining the direct effect of ChR2 opening to the top cortical layers (*Figure 2B*). This number, obtained using a Monte Carlo simulation of photon migration in tissue (*Doronin and Meglinski, 2011*, *2012*), is consistent with previous experimental and theoretical findings (*Yona et al., 2016*; *Aravanis et al., 2007*; *Foutz et al., 2012*).

FITC-labeled dextran was injected intravenously to visualize the vasculature. We measured 93 diving arterioles within the forepaw area of SI at different depths: from 30 to 560 µm below the cortical surface; 217 measured locations in total. At some locations, lateral branches were captured in the same focal plane within <200 µm lateral distance from the trunk; 88 branch measurements in total. Sensory stimulation – a 2-s train of electrical pulses (100 µs, 1 mA) delivered to a forepaw at 3 Hz – was presented at one or more measurement locations to control for normal functional hyperemia.

As with the sensory stimulus, the OG stimulus elicited a biphasic vascular response: dilation followed by constriction before returning to the baseline (*Figure 2C–D*). When measured at the same vessel location within the center of the forepaw area, the OG response featured a more pronounced constriction phase compared to the sensory one (*Figure 2D*). The overall shape of the OG response was only weakly sensitive to variation in the light pulse duration (200–430 ms) and intensity (0.7–2.8 mW) (*Figure 2—figure supplement 2A–B*). Thereafter, unless indicated, the stimulus consisted of a pair of light pulses separated by 130 ms for 450-ms total stimulus duration. These stimulus parameters produced a robust response while allowing a measurement point in between the two pulses. Averaged within a subject, 16 out of 17 OG time courses exhibited a clear constriction phase (*Figure 2E*). Grouping dilation time-courses by depth revealed that the fastest dilation onset and time-to-peak occurred at the deepest measurement locations (*Figure 2F* and *Figure 2—figure supplement 2C*). Indeed, the onset and time-to-peak gradually decreased with depth (p=1.2 $\times$ 10$^{-3}$

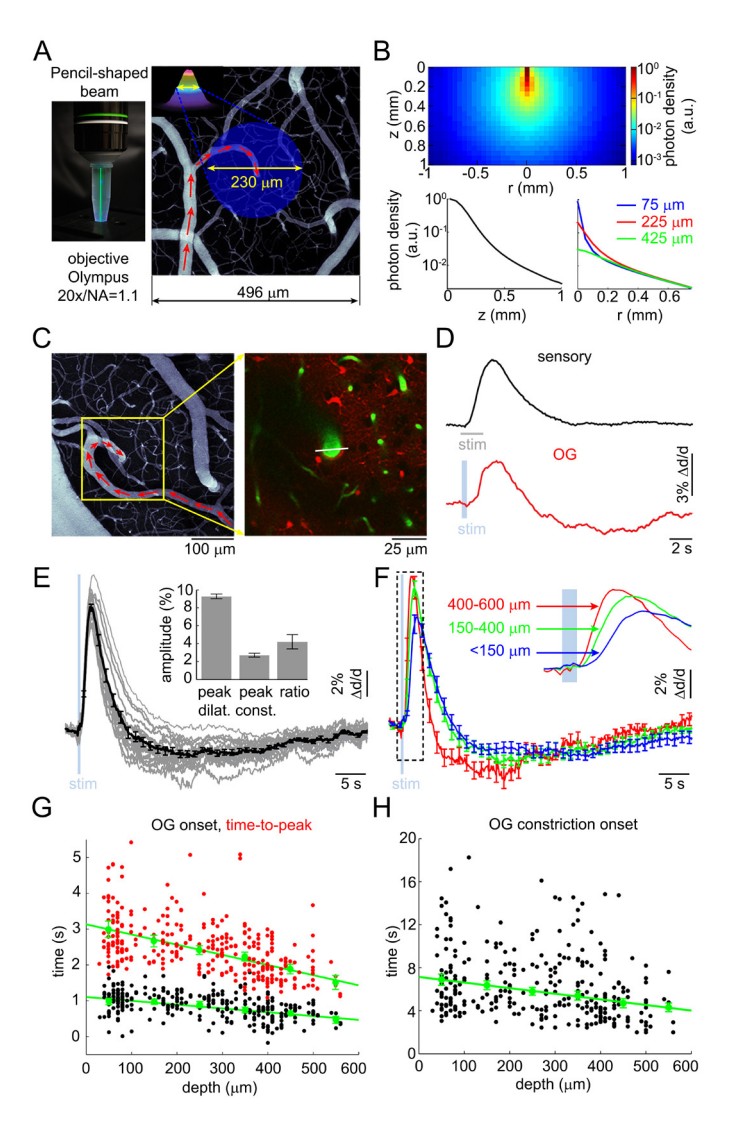

**Figure 2.** Arteriolar response to OG stimulation of INs. (**A**) Left: The 473-nm laser beam visualized in fluorescent medium. Right: Schematic illustration of the OG beam centered on a diving arteriole. The full width at half maximum (FWHM = 230 µm) of the beam is superimposed on a mean intensity projection (MIP) of a 2-photon image stack of FITC-labeled vasculature through the top 116 µm. Individual images were acquired every 3 µm. Red arrows indicate the direction of flow in the arteriole. (**B**) Simulated spatial profile of the OG beam in cortical tissue. Top: Color-coded photon density. Bottom: Photon density as a function of depth (z-axis) and as a function of the radial distance (r) at three different depths (75, 225, and 425 µm). (**C**) Left: An example vascular MIP throughout the top 180 µm. Right: the measurement plane 180 µm deep including intravascular FITC (green) and SR101-labeled astrocytes (red). The white line indicates the scanning trajectory used for diameter measurements in (**D**). (**D**) Diameter change time-courses of the diving arteriole in (**C**) in response to the sensory and OG stimuli (sensory: black, average of 10 stimulus trials; OG: red, single trial). (**E**) Each thin gray line shows an average response within one subject. Across-subject average is overlaid in thick black. Error bars indicate SE across subjects. The mean peak dilation, peak constriction, and the ratio of peak dilation to peak constriction are shown in the inset. Error bars indicate SE across subjects. (**F**) Averaged dilation time-courses grouped by depth. An expanded view of the initial 4 s after the stimulus onset is shown. The depth in µm is indicated on the left. Error bars indicate SE across subjects. (**G**) Dilation onset (black) and time-to-peak (red) as a function of depth. Conventions are as in *Figure 1C*. (**H**) As in (**G**) for constriction onset (see text).

The following figure supplements are available for figure 2:

**Figure supplement 1.** GAD67/ChR2-EYFP immunostaining of the cortex from a VGAT-ChR2(H134R)-EYFP mouse.

**Figure supplement 2.** Arteriolar response to OG stimulation of INs – additional quantification.

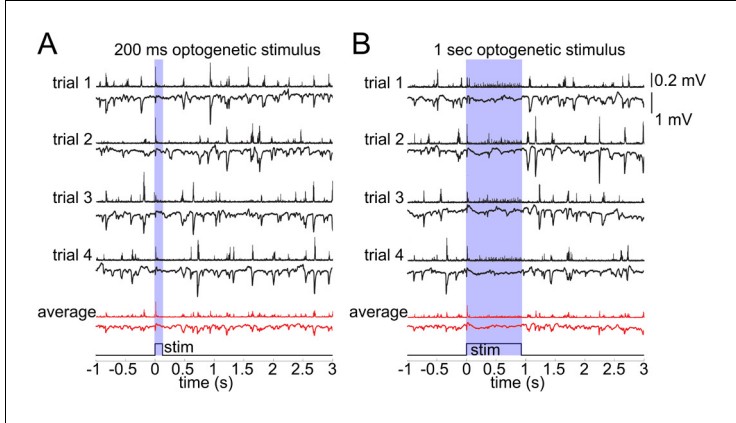

**Figure 3.** Lack of excitatory recruitment upon OG stimulation of INs. (**A**) Top: Corresponding MUA (top trace) and LFP (bottom tarce) recorded from layer II/III during a 200 ms OG stimulus. Each trace shows a single stimulus trial; red traces show the average of four individual trials. (**B**) As in (**A**) for a 1 s long OG stimulus.

The following figure supplements are available for figure 3:

**Figure supplement 1.** Lack of excitatory recruitment upon OG stimulation of INs – additional evidence from calcium imaging.

**Figure supplement 2.** Lack of excitatory recruitment upon OG stimulation of INs – additional pharmacological evidence.

and $4 \times 10^{-5}$, respectively; *Figure 2G*). The linear regression slope for the onset was consistent with dilation propagating upstream along diving arteriolar trunks at ~900 µm/s.

Next, we quantified the 'constriction onset' as the time of transition from the dilatory to constrictive phase for each measurement location. Constriction onset also gradually decreased with depth ($p=6.4 \times 10^{-6}$) indicating that deeper dilation had faster decay succeeded by constriction (*Figure 2H*). Deeper measurements exhibited a trend towards higher peak amplitude (*Figure 2—figure supplement 2D*). This trend, however, was not statistically significant due to variation between individual measurements. Finally, presenting the same OG stimulus in wild type animals did not cause any detectable physiological changes (not shown), arguing against potential dilatory/constrictive effects of heat generated by the 473-nm laser within our range of the laser power (*Christie et al., 2012*).

Taken together, these results show that selective stimulation of INs was able to reproduce the biphasic dilation/constriction sequence characteristic of sensory-induced responses. As with the sensory responses, the initial OG-induced dilation occurred in deep cortical layers where it also had the fastest rise and fastest decay superseded by constriction.

## Vascular response to the OG stimulation of INs does not rely on excitatory cells

In general, an increase in inhibition is expected to hyperpolarize postsynaptic cells reducing their excitability. Hypothetically, however, indirect activation may be produced through disinhibition or generation of a 'rebound' action potential upon termination of the hyperpolarization (*Bean, 2007*). This rebound mechanism is particularly relevant for deep PCs endowed with low threshold calcium conductance (*de la Peña and Geijo-Barrientos, 1996*). To control for these or any other unforeseen effects of the OG stimulation on PC activity, we performed extracellular recordings of the Local Field Potential (LFP) and Multi Unit Activity (MUA) in an additional 3 VGAT-ChR2(H134R)-EYFP subjects (*Figure 3*). LFP in the cerebral cortex largely reflects flow of currents along the vertically aligned PCs' apical dendrites and, therefore, provides a reliable measure of PC activity (*Buzsáki et al., 2012*; *Einevoll et al., 2013*). Our measurements revealed no LFP response timed to the termination of the OG stimulus arguing against the presence of rebound spikes in PCs (*Figure 3A*). In addition,

the ongoing (spontaneous) LFP activity was suppressed during the OG stimulus implying the hyper-polarization of PCs. The suppression effect was more obvious during longer OG stimuli (*Figure 3B*).

The MUA signal reflects spiking of all neurons, excitatory and inhibitory, within ~100 μm of the electrode tip (*Buzsáki, 2004*; *Einevoll et al., 2012*; *Pettersen and Einevoll, 2008*). During the ongoing activity, bursts of MUA coincided with downward deflections in LFP indicating participation of PCs in these spontaneous events. The OG stimulus elicited a sharp and transient increase in MUA timed to the stimulus onset (*Figure 3*). During prolonged OG stimulation, the initial transient MUA response was followed by a desynchronized activity, starting ~200 ms after the stimulus onset and lasting for the duration of the stimulus (*Figure 3B*). This MUA response in the presence of LFP suppression suggests firing of INs rather than PCs.

To provide further evidence that OG stimulation did not engage PCs, we performed in vivo calcium imaging in an additional 3 VGAT-ChR2(H134R)-EYFP subjects (*Figure 3—figure supplement 1*). PCs and INs were identified using structural reference images (*Figure 3—figure supplement 1A–B*, see Materials and methods). During spontaneous activity, the majority of imaged PCs exhibited characteristic calcium transients known to be associated with action potentials (AP) (*Figure 3—figure supplement 1C*, left) (*Grewe et al., 2010*; *Pnevmatikakis et al., 2016*; *Yaksi and Friedrich, 2006*; *Stosiek et al., 2003*). In contrast, calcium transients in INs had variable kinetics and/or small amplitude (*Figure 3—figure supplement 1C*, right). Poor quality of calcium signals in INs is consistent with previous reports and is likely to reflect low density of voltage-gated calcium channels and/or higher calcium buffering capacity in INs (*Langer and Helmchen, 2012*; *Kerlin et al., 2010*). Therefore, calcium imaging in INs may not provide enough sensitivity for the detection of single (or a few) APs generated by the 5-ms OG stimulus. Indeed, the stimulus produced no detectable calcium increase (*Figure 3—figure supplement 1C*). To ensure that this stimulus was sufficient to elicit firing, we recorded MUA activity right after acquisition of calcium imaging data. Despite the absence of a calcium increase, there was a robust MUA response to every light pulse (*Figure 3—figure supplement 1D*). While the calcium imaging experiments failed to detect firing of INs, they provided further evidence that PCs were not activated by OG stimulation. This is because no calcium increase was detected in PCs in response to the OG stimulus, although spontaneous calcium transients were readily detectable.

Taken together, these data demonstrate that indirect recruitment of PCs was unlikely to contribute to the OG-induced vascular responses. Furthermore, both the dilation and constriction phases of the response were present after blocking glutamatergic synaptic transmission with AP5 (NMDA receptor antagonist, 500 μM) and CNQX (AMPA/kainate receptor antagonist, 200 μM) (*Figure 3—figure supplement 2*, see Materials and methods). These results are in agreement with a recent laser speckle contrast imaging study in the same VGAT-ChR2(H134R)-EYFP mouse line, which concluded that the effect of OG stimulation on CBF was independent on synaptic transmission (*Anenberg et al., 2015*). Thus, arteriolar dilation and constriction in response to the OG stimulation was a consequence of direct activation of INs expressing ChR2 with no reliance on excitatory cells.

## Neurovascular mechanism induced by the OG stimulation of INs requires spiking

At first glance, the finding that OG-induced dilation initiated below layer IV (*Figure 2F–G*) while the 473-nm light used to deliver the OG stimulus was practically confined to the top 200 μm is puzzling (*Figure 2B*) (*Aravanis et al., 2007*; *Foutz et al., 2012*). However, a previous study using OG stimulation of layer V PCs has shown that stimulation at the cortical surface was effective in inducing firing of cell bodies located in layer V (*Beltramo et al., 2013*). Similar to PCs, some subtypes of INs (with bipolar, bitufted, double-bouquet, Martinotti-like morphologies [*Markram et al., 2004*]) span across the cortical depth with the top part of their dendritic and/or axonal arbor residing in the upper cortical layers. Therefore, we hypothesized that opening of ChR2 channels in these superficial neuronal processes could be sufficient to cause depolarization throughout the cell triggering release of vasoactive agents below the penetration limit of blue light in tissue, either from axonal terminals or via dendritic exocytosis (*Merighi et al., 2011*).

The superficial processes undergoing depolarization due to opening of ChR2 could be axons, dendrites, or both. Structural 2-photon imaging in the VGAT-ChR2(H134R)-EYFP mouse showed that EYFP (and, therefore, ChR2) was present in the somas and processes of INs across cortical layers (*Figure 4A*). Each EYFP-positive neuron had a number of processes emanating from the cell body

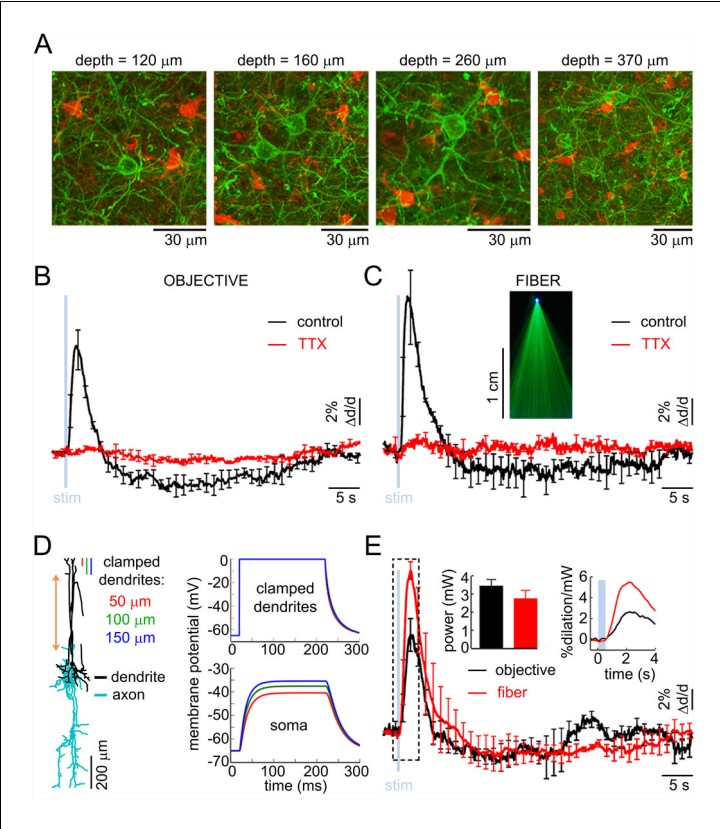

**Figure 4.** The effect of OG stimulation of INs delivered directly to layer V. (**A**) Representative images of EYFP-expressing INs at different depths (green – INs labeled with EYFP, red – astrocytes labeled with SR101). (**B**) Arteriolar response to OG stimulation of INs through the objective before and after TTX (black and red curves, respectively). Error bars indicate SE across subjects. (**C**) As in (**B**) for OG stimulation through the optical fiber with the tip in layer V. The inset shows the light-emitting fiber tip visualized in fluorescent medium. (**D**) Left: Example morphology of VIP-positive inhibitory neuron (ID=NMO_06142, NeuroMorpho.org). The red double ended arrow indicates the dendritic section that was stretched to approximate layer V neuron (see Materials and methods). Right: dendritic (top) and somatic (bottom) membrane potential in response to voltage clamping the top 50-, 100- or 150 μm of the dendritic tree at 0 mV. (**E**) Comparison of the arteriolar response (across-subject average) to OG stimulation of INs through the objective (black) and fiber (red). Error bars represent SE across subjects. The inset shows the mean laser power for each category (left) and a zoomed-in view onto the initial 4 s after the stimulus onset of the response normalized by the laser power (right).

The following figure supplements are available for figure 4:

**Figure supplement 1.** Simulation of the somatic membrane potential – another example.

**Figure supplement 2.** Approximation of neuronal morphologies.

consistent with the presence of ChR2 in the dendrites. Axonal labeling, however, could not be determined from these data.

First, we examined the possibility of ChR2 opening in the superficial axons. Assuming that ChR2 was indeed present in the axons, we reasoned that depolarization of these axons could be sufficient for triggering antidromic APs (*Schoenenberger et al., 2008*) coupling the surface illumination to the release of vasoactive messengers in deep layers. In this case, we predicted that the vasoactive effect would be abolished by blocking APs and rescued by delivering the OG light stimulus directly to layer V (i.e., bypassing the need for antidromic APs as the means of communication). To test this hypothesis, we topically applied 50 μM of tetrodotoxin (TTX), a blocker of $Na^+$ channels required for generation and propagation of APs (n = 3 subjects). TTX abolished both the dilation and constriction

response phase at all depths (*Figure 4B*). Next, we used a tapered optical fiber (see Materials and methods) positioned with the light-emitting tip ~600 µm below the surface (layer V). Under control conditions, OG stimulation through the fiber induced the characteristic biphasic arteriolar response (*Figure 4C*, black curve). In the presence of TTX, however, both the dilation and constriction phase were lost (n = 3 subjects) (*Figure 4C*, red curve).

The failure to rescue the response in the presence of TTX while delivering light directly to layer V argues against antidromic APs as the means of communication between the surface excitation and deep dilation onset. These results also imply that, in the absence of spikes, dendritic depolarization was insufficient to induce vascular responses arguing for release of vasoactive agents from axonal terminals rather than via dendritic exocytosis. In addition, these results indicate that the density of ChR2 in axonal terminals was insufficient to drive the release directly, i.e., due to the depolarization and calcium influx resulting from opening of ChR2 in these terminals. Thus, (1) spikes were required to enable release of vasoactive agents, and (2) the communication of the surface excitation to deep release likely occurred via propagation of depolarization along dendrites rather than axons.

## The dilation phase is likely to be mediated by bipolar INs with superficial dendrites and deep axons

In contrast to axons that support regenerative propagation of APs, dendritic depolarization decays with distance. Therefore, first we asked whether depolarization of superficial dendrites in INs could, in principle, drive the soma, located in layer V, above its firing threshold. To address this question, we computed membrane potential in morphologically reconstructed INs. Specifically, we chose bipolar INs positive for Vasoactive Intestinal Peptide (VIP). This choice was motivated by the desired orientation of the dendritic and axonal arbors in VIP-positive INs (*Kawaguchi and Kubota, 1996*) (*Figure 4D* and *Figure 4—figure supplement 1*) as well as the known vasodilatory properties of VIP (*Cauli et al., 2004*, *2014*).

Reconstructed VIP-positive neurons were obtained from the NeuroMorpho.Org (*Ascoli et al., 2007*) (see Materials and methods). We computed the membrane potential at the soma in response to a step depolarization ('voltage clamp') of the upper part of their dendritic tree. All computations were carried out in the Neuron simulation environment (*Hines and Carnevale, 1997*). We assumed passive membrane properties and clamped the voltage of the top 50-, 100- or 150-µm slab of the dendritic tree to 0 mV, which is the reversal potential of ChR2 (*Nagel et al., 2003*). For all three conditions, we observed a strong depolarization of the soma that developed within ~50 ms. The steady state depolarization reached values between −40 to −50 mV, depending on the number of dendritic branches within the clamped region, comparable to the reported −43.7 mV average firing threshold for VIP-positive neurons (*Neske et al., 2015*) (*Figure 4D*). In reality, dendritic membranes of INs possess active ionic conductances (reviewed in [*Cauli et al., 2014*]) enhancing the coupling of dendritic excitation to generation of spikes. Therefore, our calculation underestimates the achievable somatic depolarization. Thus, we conclude that depolarization of the upper dendrites in bipolar INs can drive spikes in somas located as deep as layer V.

The need to overcome the decay of depolarization along the dendrite implies that OG stimulation delivered directly to layer V should be more effective in inducing the vascular response compared to stimulation from the surface. To test this hypothesis, we made paired measurements in response to stimulation through the objective and fiber (positioned with the light-emitting tip ~600 µm below the surface) at each measured location (n = 30 locations along 4 arterioles in an additional 3 subjects, within 50–500 µm depth range). The stimulus consisted of a single light pulse (330–430 ms duration). *Figure 4E* shows the vascular responses, averaged across subjects, for each condition and the corresponding laser power. Normalized by the laser power, the peak dilation in response to the fiber was roughly twice as large as that in response to stimulation through the objective (*Figure 4E*, inset). Thus, initiation of dilation below layer IV in response to OG light at the cortical surface could be explained by depolarization of the superficial dendrites leading to firing of the soma (situated as deep as in layer V) and release of vasoactive agents from deep axons.

## The constriction phase is mediated by Y1 receptors for Neuropeptide Y

During the hemodynamic response to a sensory stimulus, the relative strength of arteriolar constriction in SI covaries with that of neuronal inhibition (*Devor et al., 2007*, *2008*). Thus, neurovascular

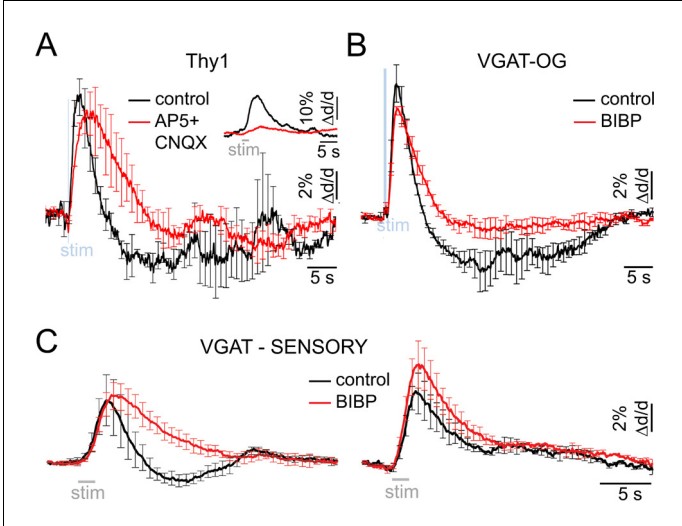

**Figure 5.** Neurovascular mechanism of the constriction phase. (**A**) Arteriolar response to OG stimulation of PCs in Thy1-ChR2-YFP subjects before (black) and after (red) application of AMPA/NMDA glutamatergic blockers (black and red, respectively). Error bars represent SE across subjects. The inset shows the sensory response before and after drug application (500 µM AP5 + 200 µM CNQX). (**B**) Comparison of dilation time-courses in response to OG stimulation of INs in VGAT-ChR2(H134R)-EYFP subjects before (black) and after (red) blocking Y1 receptors for NPY with topical application of 100 µM of BIBP 3226. Error bars represent SE across subjects. (**C**) As in (**B**) for the sensory response. The data were grouped according to the presence of the constriction phase as in *Figure 1B*.

mechanism(s) governing arteriolar constriction may be specific to activation of INs rather than excitatory neurons. To test this hypothesis, we employed Thy1-ChR2-YFP (line 18) mice where expression of ChR2 is limited to layer V PCs (*Arenkiel, 2007*). Excitation of PCs can engage other cell types because glutamate released from PCs causes depolarization and firing of postsynaptic neurons. For that reason, we applied blockers of glutamatergic synaptic transmission (AP5 and CNQX, see Materials and methods) to avoid indirect activation of neuronal cell types other than PCs. Under control conditions, OG stimulation in Thy1-ChR2-YFP mice (using a single light pulse of 50–80 ms duration) elicited a biphasic arteriolar diameter change consisting of the initial dilation followed by constriction (n = 2 subjects, 28 measurement locations along 13 arterioles within 40–380 µm depth range; *Figure 5A*, black). However, only dilation was observed in the presence of the glutamatergic blockers (*Figure 5A*, red). Thus, indirect activation of postsynaptic neurons accounted for the constriction phase. Taken together with the VGAT-ChR2(H134R)-EYFP data, these results indicate that both INs and PCs could drive dilation while constriction was selective to activation of INs.

Which subtype of INs mediates the constriction phase? Previous studies have demonstrated constriction of cortical arterioles following activation of NPY-positive INs in brain slices and during perfusion of the slice chamber (or isolated vessels) with NPY (*Cauli et al., 2004*; *Perrenoud et al., 2012*; *Dacey et al., 1988*). Further, NPY-Y1 receptors are known to be expressed by cortical microarterioles (*Abounader et al., 1999*; *Bao et al., 1997*). Motivated by these reports, we tested the effect of a Y1 antagonist BIBP 3226 on the constriction phase of vascular responses elicited by the OG stimulation in 3 VGAT-ChR2(H134R)-EYFP subjects. Topical application of 100 µM of the antagonist largely abolished the constriction phase of the OG response without a significant effect on the peak dilation amplitude (n = 52 locations along 25 diving arterioles at depths from 50 to 590 µm) (*Figure 5B*).

Next, we tested the effect of the same pharmacological treatment on dilation in response to the sensory stimulus. Sensory stimulus-induced dilation was measured at 10 locations along 7 diving arterioles in 4 additional wild type subjects at depths from 130 to 490 µm. At each location, measurements were performed before and after the topical application of 100 µM of BIBP 3226. *Figure 5C* shows the averaged time-courses before and after BIBP 3226 (black and red, respectively). For vessels exhibiting a clear biphasic response, the constriction phase was abolished by BIBP

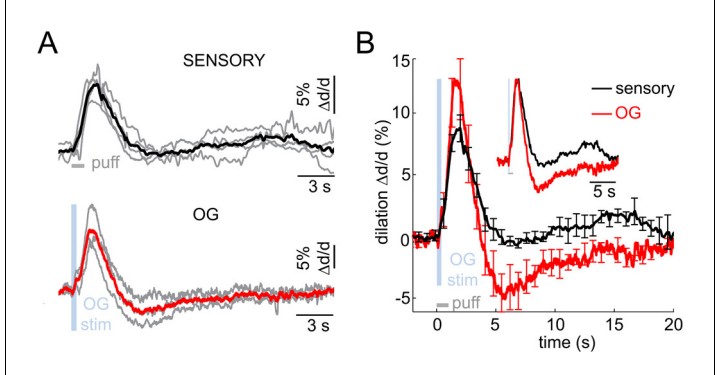

**Figure 6.** Imaging arteriolar response to sensory and OG stimulation of INs in awake mice. (**A**) Arteriolar dilation in awake mice in response to a sensory stimulus (three 100-ms air puffs to the contralateral whisker pad, top panel) and OG stimulation of INs (bottom panel). Thin gray lines represent individual subjects. Across-subject averages are overlaid (thick black and red lines for the sensory and OG stimuli, respectively). (**B**) Overlaid dilation time-courses in response to sensory stimulation (black) and OG stimulation of INs (red). Error bars represent SE across subjects. The inset shows the same time-courses normalized to the peak.

3226 without a significant effect on the peak dilation amplitude (*Figure 5C*, left). For vessels exhibiting monophasic dilation, there was a trend towards higher dilation amplitude after BIBP 3226 (*Figure 5C*, right). This trend, however, was not significant.

These results are consistent with prior demonstrations in brain slices that dilation and constriction are mediated by different types of INs (*Cauli and Hamel, 2010*; *Cauli et al., 2004*; *Perrenoud et al., 2012*). We conclude that the constrictive effect elicited by both the OG and sensory stimuli was driven by NPY – presumably released from NPY-positive INs (*Karagiannis et al., 2009*) – acting on Y1 receptors.

## Biphasic arteriolar response to sensory and OG stimuli in the absence of anesthesia

Anesthesia can differentially affect neuronal cell types altering neuronal network activity and neurovascular coupling. To this end, we performed two-photon imaging of single diving arterioles in awake mice, i.e., in the absence of general anesthesia or any sedative drugs (see Materials and methods). The sensory stimulus consisted of three gentle air puffs onto the whisker pad (100-ms puffs delivered at 3 Hz). Diving arterioles within the whisker pad representation of SI (i.e., Barrel cortex) displayed readily detectable biphasic arteriolar responses to the sensory stimuli consisting of dilation followed by a smaller amplitude constriction (n = 4 subjects, 11 measurement locations along 11 arterioles within 30–120 μm depth range; *Figure 6A* top, black). The overall response duration was shorter than that in the anesthetized subjects (*Figure 1B*), most likely due to differences in the stimulus duration and strength (1-s train of air puffs in awake vs. 2-s train of electrical pulses under anesthesia). In addition, awake subjects exhibited a secondary dilation of smaller amplitude 10–15 s after the stimulus onset, likely reflecting cognitive processing and, in some cases, active whisking following the sensory stimulus.

Next, we performed measurements in awake VGAT-ChR2(H134R)-EYFP subjects in response to OG stimulation of INs (n = 3 subjects, 11 measurement locations along 10 arterioles within 30–120 μm depth range; *Figure 6A* bottom, red). The stimulus consisted of a single light pulse (150–400 ms duration). Similar to the response under anesthesia, the OG stimulus induced biphasic arteriolar diameter changes with more pronounced constriction compared to the sensory response (*Figure 6B*). These results rule out the possibility that the constriction phase could be an epiphenomenon of anesthesia establishing the biphasic dilation/constriction sequence as a normal feature of the hemodynamic response at the level of parenchymal arterioles.

## Discussion

A key goal in neurovascular research is establishing molecular mechanisms mediating dilation and constriction of the brain microvasculature driven by changes in neuronal activity. In the present study, we used optogenetics to isolate the mechanisms originating from the inhibitory (GABAergic) population of cortical neurons. We demonstrate that OG stimulation elicited a biphasic diameter change composed of a combination of dilatory and constrictive phases. While dilation was also induced by OG stimulation of PCs, the constriction response was specific to activation of INs. Further, we identified the vasoconstrictive molecular agent as NPY acting on Y1 receptors and showed that the same NPY-Y1 pathway was responsible for the sensory stimulus-induced vasoconstriction. This finding supports previous observations in brain slices implicating NPY-positive INs as the primary mediators of vasoconstriction in cortical arterioles under normal conditions (*Cauli et al., 2004*; *Perrenoud et al., 2012*).

Previously, we reported the center-surround and contra-ipsilateral patterns of arteriolar diameter changes in SI in response to a sensory stimulus that were dominated by dilation within the receptive field and constriction in the surrounding region as well as in the ipsilateral to the stimulus hemisphere (*Devor et al., 2005*, *2007*, *2008*). The strength of constriction in these studies correlated with the strength of inhibition. In contrast to natural behavior of cortical circuits where specific types of INs play a differential role in 'local' and 'lateral' inhibition (*Helmstaedter et al., 2009*), OG stimulation of INs in the present study produced indiscriminate activation across various inhibitory cell types. Therefore, arteriolar responses to the OG stimulation cannot be directly compared to the results from these previous studies. However, the finding of the NPY-Y1 pathway underlying constriction predicts that NPY-positive INs may be specifically involved in shaping the surround and ipsilateral neuronal inhibition.

OG stimulation in the VGAT-ChR2(H134R)-EYFP line produced a monophasic CBF increase in a recent laser speckle contrast imaging study by Anenberg et al. (*Anenberg et al., 2015*). Arteriolar constriction, however, would normally result in a CBF decrease, as we demonstrated previously (*Devor et al., 2007*, *2008*). We speculate that the high-frequency stimulation protocol used by Anenberg and colleagues may have contributed to the lack of constriction and CBF decrease in their study. In contrast to other types of INs driving dilation, at least some NPY-positive neurons exhibit a pronounced spike latency (*Karagiannis et al., 2009*) and thus may not respond well to short (5-ms duration) light pulses delivered at high frequency (100 Hz).

Prior studies on neuropeptides in different brain areas have demonstrated that the release could occur from axonal terminals as well as via dendritic exocytosis (*Merighi et al., 2011*). In the present study, OG stimulation of ChR2-expressing dendrites and somas under TTX was not effective in causing vascular responses while delivering the stimulating light either to the cortical surface or deep layers. These results suggest that the corresponding vasoactive agents, including NPY, were released from axons rather than via dendritic exocytosis, while spikes were required to drive the release. Yet, we cannot rule out a possibility of dendritic release induced by back-propagating APs (*Stuart and Sakmann, 1994*). Indeed, back-propagating APs have been reported in bipolar, bitufted and neurogliaform INs (*Goldberg et al., 2003*; *Kaiser et al., 2001*; *Cho et al., 2010*).

The requirement for spiking during the OG response should not be interpreted in the context of the 'classical' question of whether it is the spiking or synaptic activity that correlates better with BOLD fMRI. Historically, this question was motivated by the idea of a metabolic feedback, postulating that CBF increase was mechanistically related to the accumulation of vasoactive energy metabolites (*Raichle and Mintun, 2006*). The present results, on the other hand, demonstrate a feed forward mechanism where specific vasoactive signaling agents released by active neurons drive dilation and constriction, depending on neuronal cell type. Further, the initial dilation in our experiments occurred in infragranular layers and not in layer IV, which is the most metabolic layer in SI (*Sokoloff et al., 1977*). These findings do not contradict the fact that some of the energy metabolites are vasoactive (*Gordon et al., 2008*). Rather, they suggest that, under healthy conditions in vivo it is the vasoactive messengers related to neuronal signaling and not energy metabolites that play the dominant role in CBF regulation.

The identity of vasodilatory mechanisms induced by OG stimulation of INs requires further investigation. Previous studies put forward a number of possibilities including release of VIP and NO (*Cauli and Hamel, 2010*; *Liu et al., 2012*). The present results indicate that INs mediating dilation

may have bipolar morphologies with dendrites extending to the surface and axons reaching to deep cortical layers. VIP-positive INs meet these morphological criteria, and some of them co-express neuronal nitric oxide synthase (*Kawaguchi and Kubota, 1996*; *Cauli et al., 2014*; *Perrenoud et al., 2012*). These features as well as known vasodilatory properties of VIP (*Cauli et al., 2004*, *2014*) put forward VIP-positive INs as a primary suspect initiating dilation in deep layers. Future experiments using restricted expression of OG actuators/inhibitors in sub-populations of INs with known repertoire of vasoactive signaling molecules would be required to comprehensively address this hypothesis.

It is well established that dilation and constriction can propagate along arteriolar walls (*Segal and Duling, 1986*; *Rosenblum et al., 1990*; *Jensen and Holstein-Rathlou, 2013*; *Chen et al., 2014*; *Peppiatt et al., 2006*; *Welsh and Segal, 1998*), and these conducted responses may have contributed to the spread of OG dilation and constriction throughout the cortical depth in our data. This possibility is consistent with the observed slowing down of dilation kinetics towards the cortical surface and the trend towards a gradual decrease in the peak amplitude – this type of behavior is expected for a decaying propagated response (*Iadecola et al., 1997*; *Chen et al., 2011*). These findings, however, do not imply that neurovascular coupling mechanisms reside exclusively in deep cortical layers and do not rule out local lamina-specific neurovascular communication at every depth. In the presence of both local and conducted signaling, the onset of dilation would be determined by the faster of the two processes. Therefore, substantial delays in the superficial arteriolar branches suggest that local neurovascular communication in the upper layers, if it exists, has slower kinetics.

The observed gradual decrease of the dilation latency and time-to-peak down to the infragranular layers is in agreement with laminar resolved functional Magnetic Resonance Imaging (fMRI) in human (*Siero et al., 2011*) but at odds with a recent high resolution fMRI in the rat that used a line-scanning method (*Yu et al., 2014*). The apparent discrepancy with *Yu et al. (2014)* may reflect the complex nature of the Blood Oxygenation Level Dependent (BOLD) fMRI signal that depends on the balance between $O_2$ delivery and consumption as well as on the measurement theory specific to the chosen data acquisition paradigm. Our present conclusions, on the other hand, rely on direct and assumption-free measurements of the physiological parameter of interest – the arteriolar diameter.

Astrocytic contribution to either dilation or constriction evoked by the OG stimulation of INs was not specifically addressed in the current study. A number of recent reports including our own have provided negative evidence for the involvement of calcium-dependent release of vasoactive gliotransmitters in normal regulation of cortical blood flow by local neuronal activity (*Nizar et al., 2013*; *Bonder and McCarthy, 2014*; *Takata et al., 2013*; *Jego et al., 2014*). However, we cannot rule out calcium-independent pathways. Expression of Y1 receptors has been documented in astrocytic cultures (*Abounader et al., 1999*); although, more recent transcriptomic analyses of astrocytes isolated from adult mouse brains have revealed little or no expression of these receptors (*Tasic et al., 2016*; *Cahoy et al., 2008*).

The present study has a number of limitations. As such, the pattern of OG stimulation in space and time did not mimic naturally occurring neuronal activity. In addition, the possible variation in the level of ChR2 expression could have resulted in bias towards specific INs cell types. The design of our OG study, however, did not rely on natural or balanced neuronal activity patterns. Instead, our major goal was to produce artificially selective activation of cortical INs as a single population for isolation of their vasoactive effects. Another potential pitfall is that OG stimulation of INs could lead to an indirect recruitment of PCs, which have been implicated in regulation of blood flow (*Lecrux et al., 2011*). However, our electrophysiological recordings and calcium imaging data produced no evidence for PC recruitment. Finally, the majority of the experiments in our study were performed under anesthesia and in the presence of a paralytic agent that could have affected the vasoreactivity and neurovascular coupling. As such, the ketamine/xylazine anesthesia employed by Anenberg et al. (*Anenberg et al., 2015*) may provide an additional explanation for the discrepancy in our results regarding the constriction phase in response to OG stimulation of INs. Stepping away from anesthesia is an ongoing effort in many laboratories, including our own, but still remains a challenge for studies that require pharmacological manipulations, insertion of recording electrodes, and deep imaging. In the present study, we have confirmed the biphasic nature of the arteriolar diameter change in awake mice, both in response to a sensory stimulus and OG activation of INs.

The constrictive effect of INs demonstrated in the present study, taken together with previous theoretical calculations suggesting that inhibition has lower metabolic costs than excitation

(*Attwell and Laughlin, 2001*, *2002*), raises the possibility that paired CBF and cerebral metabolic rate of $O_2$ (CMRO$_2$) measurements can provide information about the net respective contributions of excitatory and inhibitory activity within the ensemble neuronal network response, due to their differential vasoactive role and energetic costs. This possibility is of particular interest for the quantitative BOLD fMRI approach (a.k.a. the "calibrated BOLD") that combines the BOLD and arterial spin labeling (ASL) methods to isolate the effects of CBF and CMRO$_2$ (*Davis et al., 1998*; *Hoge, 2012*; *Pike, 2012*). Thus, we are tempted to speculate that identifying the CBF and CMRO$_2$ effects induced by excitatory and inhibitory neurons could open a new direction in which quantitative fMRI may be able to provide information on the underlying neuronal activity.

## Materials and methods

### Animal procedures for imaging under anesthesia

All experimental procedures were performed in accordance with the guidelines established by the UCSD Institutional Animal Care and Use Committee (IACUC). We used 103 adult mice of either sex including 48 VGAT-ChR2(H134R)-EYFP and 2 Thy1-ChR2-YFP (using promoters from *Slc32a1* and *Thy1* genes, respectively; Jackson Stock Numbers 014548 and 007612, respectively; both heterozygous on a mixed C57Bl6/ICR background), and 53 wild type ICR. Surgical procedures in mice expressing ChR2 were performed in a dark room using a 515 nm longpass filter (Semrock FF01-515/LP-25) in the surgical microscope light source to avoid OG stimulation during installation of the cortical window. Mice were anesthetized with isoflurane during surgical procedures (2% initially, 0.5–1% during all procedures). A cannula was inserted into the femoral artery. A metal holding bar was glued to the temporal bone for immobilization of the head during imaging. An area of skull overlying the forepaw region of the primary somatosensory cortex (SI) contralateral to the holding bar was exposed and dura mater removed. A ~2 × 2 mm cranial window was centered on stereotactic coordinates: AP -0.5, ML 2.25.

In the majority of experiments, the red fluorescent dye sulforhodamine 101 (SR101, S7635, Sigma) in artificial CSF (ACSF) was applied topically for ~2 min to label astrocytes (*Nimmerjahn et al., 2004*) providing a contrast in tissue that was used for visual assessment of potential damage due to experimental procedures. ACSF contained 142 mM NaCl, 5 mM KCl, 10 mM glucose, 10 mM HEPES, 3.1 mM CaCl$_2$, 1.3 mM MgCl$_2$, pH 7.4. The excess dye was washed with ACSF. A drop of agarose (1% wt/vol, A9793, Sigma) in ACSF was applied on the brain surface, and the exposure was covered with a round glass coverslip (5 mm, WPI) and sealed with dental acrylic. To avoid herniation of the exposed brain due to excessive intracranial pressure, the dura mater over the IVth cerebral ventricle was punctured, thus allowing drainage of CSF. After the exposure was closed, the drainage hole was sealed with agarose.

In experiments involving pharmacological manipulations, calcium imaging, or insertion of an optical fiber, the round glass coverslip was cut straight on one side facing a gap in the dental acrylic seal. The agar was cut down along the cut side forming a vertical wall. The exposure was aligned with the agar wall such that ACSF under the objective was in direct contact with the cortical surface allowing drugs to penetrate into the cortical tissue.

After closing of the exposure, mice were left to rest under 1% isoflurane for 45 min. Then, isoflurane was discontinued and anesthesia maintained with α-chloralose (50 mg/kg/h IV, C0128, Sigma or 100459, MP Biochemicals). Mice were paralyzed with pancuronium bromide (0.4 mg/kg/h IV, P1918, Sigma) (*Shin et al., 2007*) and ventilated (~110 min$^{-1}$) with 30% $O_2$ in air. Fluorescein isothiocyanate (FITC)-labeled dextran (MW = 2 MDa, FD-2000S, Sigma) was injected IV (50–100 µl of 5% (w/v) solution in phosphate-buffered saline) to visualize the vasculature and control for the integrity of capillary bed. Expired $CO_2$ was measured continuously using a micro-capnometer (Cl240, Columbus instruments). Heart rate, blood pressure, and body temperature were monitored continuously. Blood gas was analyzed to cross-validate the micro-capnometer measurements. Respiration was adjusted to achieve PaCO$_2$ between 30 and 40 mmHg and pH between 7.35 and 7.45. α-chloralose and pancuronium in 5% dextrose saline were supplied through the femoral line every 30 min for the duration of data acquisition. Waiting for 45 min between closing of the exposure and drug injections minimized leakage of the drugs onto the exposed cortical tissue through the cut dural blood vessels.

For calcium imaging experiments, calcium indicator Oregon Green 488 BAPTA-1 AM (OGB1) (O-6807, Invitrogen, 50 µg) was first dissolved in 4 µl of 20% pluronic in DMSO (F-127, Invitrogen); 80 µl of ACSF were added to the OGB1 solution to yield a final concentration of 0.5 mM OGB1. The microinjection pipette was guided under the glass coverslip and positioned ~ 200 µm below the cortical surface using a Luigs & Neumann translation stage (380FM-U) and manipulation equipment integrated into the Ultima system. The red fluorescent dye Alexa 594 (A-10442, Alexa Fluor 594 hydrazide, sodium salt, Invitrogen) was added to the OGB1 solution in order to visualize the micropipette during manipulation and to provide visual feedback during pressure-microinjection into the cortical tissue (*Stosiek et al., 2003*). The pressure was manually adjusted to ensure visible spread of Alexa 594 while avoiding movement artifacts.

## Animal procedures for imaging awake mice

We used a Polished, Reinforced Thinned-Skull (PoRTS) technique (*Drew et al., 2010*) for installation of chronic 'cortical windows' providing sufficient visibility for imaging of diameter changes in single diving arterioles down to ~120 µm below the surface. During the PoRTS procedure, mice were anesthetized with isoflurane (2% initially followed by 1% during the surgery); their body temperature was maintained at 37°C. A custom holding bar allowing repeated head immobilization was glued to the skull overlaying the left hemisphere. On the right side, a ~3 3 mm area of skull was thinned until translucent and polished with silicon carbide grit powder (Convington Engineering). The exposure was centered on the Barrel cortex region of SI using stereotactic coordinates: AP −1.5, ML 2. A glass coverslip was glued to the thinned skull and fixed along the perimeter with dental acrylic. Additional dental acrylic was applied around the holding bar joining to the perimeter of the coverslip in order to reinforce the overall assembly.

After surgical implantation of the bar and a full day recovery, mice were habituated in 1 session/ day to accept increasingly longer periods of head restraint under the microscope objective (up to 2 hr). During the head restraint, the animal was placed on a suspended bed. A drop of sweetened condensed milk was offered every 15 min during the fixation as a reward. Habituated head-fixed mice consumed the reward milk. They were free to readjust their body position and from time to time displayed natural grooming behavior. A video camera (Lifecam Studio, Microsoft; IR filter removed) with an NIR longpass filter (Midwest Optical LP920-25.5) was used for continuous observation of the mouse. The IR illumination (M940L3 - IR (940 nm) LED, Thorlabs) was invisible for the PMT photodetectors and generated no imaging artifacts. The camera frames were synchronized with two-photon imaging and recorded. Periods with extensive body movement (e.g., grooming behavior) were excluded during data analysis.

## Sensory stimulation

In experiments under anesthesia, sensory stimulation was delivered to the forepaw contralateral to cortical exposure through a pair of thin needles inserted under the skin using a train of six 100 µs, 1-mA electrical pulses at 3 Hz. This stimulus paradigm was chosen because it produced synchronized neuronal spiking response accompanied by robust dilation with a sharp onset (*Devor et al., 2008*; *Tian et al., 2010*; *Nizar et al., 2013*). All measurements in response to the sensory stimulus were performed within a 1-mm radius from the center of the forepaw region of SI determined by the stereotactic coordinates (AP 0.5, ML 2.25). Ten stimulus trials were averaged at each measurement location.

In awake mice, sensory stimulus consisted of a train of three air puffs onto the whiskers contralateral to the cortical window. We used three 100-ms puffs at 3 Hz delivered through a plastic tube (2 mm inner diameter). The tube was positioned behind the whiskers to minimize the eye blink reflex. Seven to nine stimulus trials were averaged at each measurement location.

Stimulation devices (A365 stimulus isolator or PV830 picopump, WPI) were triggered using a separate PC that also acquired timing signals for data acquisition ('trigger out' signals for each frame/ line) and physiological readings using a National Instruments IO DAQ interface (PCI-6229) controlled by custom-written software in MATLAB (MathWorks Inc.). The timing of each frame/line relative to the stimulus onset was determined during data analysis based on acquired triggering signals.

## OG stimulation

OG stimulation was delivered though the objective using a 473-nm cylinder-shaped laser beam ~230 µm in diameter (FWHM) (*Figure 2A–B*) that is comparable to the size of a cortical column. The spatial distribution of the photon density in tissue produced by the OG beam (*Figure 2B*) was estimated using Monte Carlo simulation of photon migration in tissue (*Doronin and Meglinski, 2011*, *2012*) assuming the following cortical gray matter optical parameters at 473 nm: absorption coefficient $\mu_a$ = 0.2 mm$^1$, scattering coefficient $\mu_s$ = 68 mm$^1$, and anisotropy factor g = 0.95 (*Flock et al., 1992*). The beam was centered on a diving arteriole using a dedicated set of galvanometer mirrors. The duration of the light pulse was controlled by a dedicated shutter and synchronized with imaging. Only a single OG trial was presented at each measurement location to avoid overstimulation.

In some experiments, the 473-nm beam was coupled to a tapered optical fiber inserted in the cortical tissue (*Pisanello et al., 2014*). Tapered optical fibers were purchased from Nanonics Imaging Ltd. (MM-UV fiber for 200–1200 nm wavelengths, core diameter 50 µm, cladding diameter 125 µm, 250 µm protective acrylate buffer coating (except 2–3 mm from the tip), core refractive index 1.464, cladding refractive index 1.447, numerical aperture 0.22, taper angle between 3° and 6°, Cr-Au reflective coating thickness 300 nm, aperture diameter at taper tip ~200 nm).

Tapering of the fiber was critical to minimize tissue damage during penetration (*Pisanello et al., 2014*). To generate a larger 'blob' of light in deep cortical layers while retaining the needle shape necessary for smooth penetration, we etched 200 µm of the coating from the tip (custom modification by Nanonics). The resultant beam profile is shown in *Figure 4C*. The fiber was guided under the glass coverslip and positioned using Luigs and Neumann translation stage (380FM-U) and manipulation equipment integrated into the Ultima system.

## Two-photon imaging

Images were obtained using an Ultima two-photon laser scanning microscopy system from Bruker Fluorescence Microscopy (formerly Prairie Technologies) equipped with an Ultra II femtosecond Ti: Sapphire laser (Coherent) tuned between 800–1000 nm. For penetration deeper than ~600 µm, an Optical Parametric Oscillator (Chameleon Compact OPO, Coherent), pumped by the same Ti:Sapphire laser, was tuned to 1360 nm. The OPO was used in conjunction with the intravascular administration of dextran-conjugated Alexa Fluor 680 (D34680, Invitrogen) (*Kobat et al., 2009*). FITC and Alexa Fluor 680 were imaged using cooled GaAsP detectors (Hamamatsu, H7422P-40). SR101 was imaged using a multialkali PMT (Hamamatsu, R3896).

In experiments involving OG stimulation, the main dichroic mirror contained a 460–480 nm notch (Chroma ZT470/561/NIR TPC). An additional filter blocking wavelengths in the range 458–473 nm (Chroma ZET458-473/561/568/NIR M) was added in front of the PMT block. Nevertheless, residual bleed-through of the 473-nm light prevented us from using GaAsP detectors. Therefore, in these experiments, FITC (or OGB1) and SR101 were imaged using a pair of multialkali PMTs.

We used a 4x objective (Olympus XLFluor4x/340, NA=0.28) to obtain low-resolution images of the exposure. Olympus 20x (XLUMPlanFLNXW, NA=1.0) and Zeiss 40x (IR-ACHROPLAN, NA = 0.8) water-immersion objectives were used for high-resolution imaging. In experiments involving manipulation of a micropipette or optical fiber under the coverslip, we used a combination of Zeiss 5x (Plan-NEOFLUAR, NA=0.16) and Olympus 20x (UMPlanFI, NA=0.5) objectives for a coarse approach and fine manipulation under the glass coverslip, respectively. The laser beam diameter was adjusted to overfill the back aperture. Diameter measurements were performed in a frame-scan mode at 10–20 Hz, or in a 'free-hand' line-scan mode with a scan rate of 25–50 Hz. The scan resolution was 0.5 µm or less. Calcium imaging was performed in line-scan mode at 100 Hz, 20–50 pixels per neuron, and ~10 µs dwell time.

## Identification of INs and PCs in vivo and combination of 2-photon calcium imaging and OG stimulation

First, we acquired two-photon Z-stacks of structural reference images. Since ChR2 in the VGAT-ChR2(H134R)-EYFP line is fused with enhanced yellow fluorescent protein (EYFP) (*Zhao et al., 2011*), we used EYFP fluorescence to identify INs. SR101 was used to label astrocytes (*Nimmerjahn et al., 2004*) (*Figure 3—figure supplement 1A*). Next, we loaded cells with OGB1 within the same volume

and performed in vivo calcium imaging in cortical layer II/III. OGB1 has a strong spectral overlap with EYFP, and, in addition, ChR2-EYFP expression in the VGAT-ChR2(H134R)-EYFP line is relatively weak. As a result, EYFP was not detectable after addition of OGB1. Therefore, we used SR101 to coregister OGB1 images with the reference (EYFP/SR101) images. EYFP-positive cells were labeled as INs, and cells negative for EYFP and SR101 were labeled as PCs (*Figure 3—figure supplement 1B*). Data could not be acquired during OG stimulation because, despite the optical filtration, some of 473 nm excitation light reached the photodetectors causing their saturation. Therefore, we used brief (5 ms) light pulses and acquired line scans at 100 Hz (10 ms per line) limiting the light artifact to 20–30 ms. In this regime, we were in a good position to detect calcium transients that have a fast rise and slow decay (time constant ~1 s [*Lütcke et al., 2013*]).

## Pharmacology

All drugs were applied topically under the objective and let to diffuse into the cortical tissue for at least 40 min. D-(-)-2-amino-5-phosphonopentanoic acid (AP5, 500 µM in ACSF, A5282, Sigma) and 6-Cyano-7-nitroquinoxaline-2,3-dione (CNQX, 200 µM in ACSF, C239, Sigma) were used to block AMPA and NMDA receptors for glutamate, respectively. Tetrodotoxin (TTX, 50 µM in ACSF, 554412, Calbiochem), was applied to block $Na^+$ channels required for generation and propagation of APs. N2-(Diphenylacetyl)-N-[(4-hydroxyphenyl)methyl]-D-arginine amide (BIBP 3226, 100 µM in ACSF, B174, Sigma) was applied to block Y1 receptors for NPY.

## Immunofluorescence staining

Brains extracted from adult VGAT-ChR2(H134R)-EYFP or wild type mice were placed in phosphate-buffered saline (PBS) with 4% paraformaldehyde for 12 hr at 4°C and then in PBS with 30% sucrose at 4°C for 24 hr. 40 µm coronal sections were cut with a vibratome (The Vibratome Company). Sections were placed in PBS with 0.5% bovine serum albumin (BSA). After permeabilization with 1% Triton X-100 and 0.5% BSA in PBS for 1 hr and blocking with 10% normal goat serum, 0.5% BSA, and 0.1% Triton X-100 in PBS for 3 hr, sections were incubated overnight at 4°C with primary antibodies (anti-GAD67, mouse monoclonal clone 1G10.2, Millipore MAB5406; anti-EGFP, rabbit polyclonal antiserum, Abcam ab290; both diluted 1:1000 in blocking solution). After washing, sections were incubated with goat anti-mouse and goat anti-rabbit antibodies coupled to Alexa 594 and Alexa 488 (Invitrogen A 11,005 and A 11034), respectively, which were diluted 1:500 in blocking solution. Sections were mounted on glass slides with ProLong Gold Antifade Mountant with DAPI (Invitrogen, Carlsbad, California). To confirm specificity of the EGFP antiserum, we stained sections from a wild type mouse. To control for unspecific binding of the secondary antibodies, we incubated sections only with the secondary antibodies. Stained sections were imaged on the 2-photon microscope described under 'Two-Photon Imaging' using 750 nm excitation, an Olympus 20x (XLUM-PlanFLNXW, NA=1.0) objective and 460/60 nm, 525/50 nm, 670/50 nm filters to detect DAPI, Alexa 488, and Alexa 594 emissions. Two-photon image stacks (Z-step size 3 µm, 1024x1024 pixel, 400x400 µm field-of-view) were used to count $GAD67^+$, $EYFP^+$, or $GAD67^+/EYFP^+$ cells.

## Extracellular electrophysiological recordings

Extracellular recordings of LFP and MUA were acquired as described in (*Devor et al., 2003*, *2005*; *Einevoll et al., 2007*), using a tungsten microelectrode (FHC, 6–8 MΩ). The recorded potential was amplified and filtered into two signals: a low-frequency part (0.1–500 Hz, sampled at 2 kHz with 16 bits) and a high-frequency part (150–5000 Hz, sampled at 20 kHz with 12 bits). The low-frequency part is referred to as the LFP. The high-frequency part was further filtered digitally between 750 and 5000 Hz using a zero phase-shift second-order Butterworth filter and rectified along the time axis to provide the MUA.

Microelectrodes were guided under the glass coverslip and positioned in cortical layer II/III using Luigs & Neumann translation stage (380FM-U) and manipulation equipment integrated into the Ultima system. In calcium imaging experiments, the electrode was moved into the field of view right after acquisition of the imaging data. Simultaneous two-photon imaging and electrophysiological recordings from exactly the same location was not possible because of the photovoltaic artifact resulting from direct exposure of the metal microelectrode to focused Ti:Sapphire laser light.

## Computing membrane potential in reconstructed neurons

Reconstructed neuronal morphologies were obtained from NeuroMorpho.Org (*Ascoli et al., 2007*). We used two VIP-positive cells (ID= NMO_06142, NMO_06144) from layer II/III of the rat Barrel cortex provided by Bruno Cauli (*Karagiannis et al., 2009*). First, the original morphologies were scaled in Z (depth axis) by 0.6 to account for differences in the cortical thickness between the rat and mouse SI. Next, we stretched the middle dendritic section from 100 µm below the surface (the border between layers I and II) to the soma along the depth axis to position the soma in layer V (600 µm deep) approximating morphology of a VIP-positive neuron from layer V (*Figure 4—figure supplement 2*). All computations were carried out in the Neuron simulation environment (*Hines and Carnevale, 1997*) assuming passive membrane properties.

## Imaging data analysis

Data were analyzed in MATLAB using custom-written software as described in (*Tian et al., 2010*; *Nizar et al., 2013*). Unless indicated, statistics were performed across subjects. In scatter plots, measurements were group-averaged according to the depth in 100-µm bins for each subject before calculating the mean and SE across subjects for each bin. p-value for the regression parameters of the scatter data, in particular, slope values for the linear regression fits, were computed using MATLAB's statistical toolbox function *regstats()*. The reported p-value is for the t statistic using the null hypothesis that the slope is equal to zero.

When multiple categories were defined (e.g., time-courses with and without constriction in *Figure 1B*), we averaged measurements for each category within a subject prior to performing statistics across subjects. When peak normalization was applied, we first averaged all time courses acquired within a subject. Then, averaged time courses were normalized by the peak amplitude before calculating the average across subjects.

For analysis of calcium imaging data, acquired in line-scan mode, line segments corresponding to individual neuronal cell bodies were segmented based on their intensity profile (cell bodies were brighter than neuropil). For each sampled neuron, the calcium signal per time point (i.e., one line of the line-scan) was calculated as an average of all pixels within the corresponding line segment. This calculation was repeated for each line in the time series to generate a single-neuron time-course. The same procedure was performed separately for each neuron, resulting in a family of neuron-specific time-courses for each line-scan. After digital removal of the OG light artifact, data were downsampled by a factor of two along the time axis to provide a 20-ms time resolution.

## Acknowledgements

We gratefully acknowledge support from the NIH (NS057198, EB00790, U01NS094232, NS082097, EB003832 and S10RR029050), the Research Council of Norway, and the Ministry of Education, Youth and Sports of the Czech Republic (project CEITEC 2020 (LQ1601)). K Kılıç was supported by a postdoctoral fellowship from the International Headache Society in 2014 and TUBITAK in 2015. F Razoux, M Thunemann and M Desjardins were supported by postdoctoral fellowships from the Swiss National Science Foundation, German Research Foundation (DFG TH 2031/1) and Natural Sciences and Engineering Research Council of Canada, respectively.

## Additional information

### Competing interests

DK: Reviewing editor, *eLife*. The other authors declare that no competing interests exist.

### Funding

| Funder | Grant reference number | Author |
| --- | --- | --- |
| European Regional Development Fund | CEITEC CZ.1.05/1.1.00/ 02.0068 | Hana Uhlirova |
| International Headache Society | | Kıvılcım Kılıç |

| | | |
|---|---|---|
| Türkiye Bilimsel ve Teknolojik Araştirma Kurumu | | Kıvılcım Kılıç |
| John Carroll University, Department of Physics | | Peifang Tian |
| Deutsche Forschungsgemeinschaft | DFG TH 2031/1 | Martin Thunemann |
| Natural Sciences and Engineering Research Council of Canada | | Michèle Desjardins |
| Schweizerischer Nationalfonds zur Förderung der Wissenschaftlichen Forschung | | Florence Razoux |
| National Institutes of Health | NS082097 | David Kleinfeld |
| National Institutes of Health | EB003832 | David Kleinfeld |
| Norges Forskningsråd | | Gaute T Einevoll |
| National Institutes of Health | EB00790 | Anders M Dale |
| National Institutes of Health | S10RR029050 | Anders M Dale |
| National Institutes of Health | NS057198 | Anna Devor |
| National Institutes of Health | U01NS094232 | Anna Devor |

The funders had no role in study design, data collection and interpretation, or the decision to submit the work for publication.

## Author contributions

HU, KK, PT, MT, MD, Final approval of the version to be published, Acquisition of data, Analysis and interpretation of data, Drafting or revising the article; PAS, SS, TVN, KN, VBS, TCS, GTE, Final approval of the version to be published, Analysis and interpretation of data, Drafting or revising the article; CM, QC, KLW, FR, MV, JAC, CGLF, Final approval of the version to be published, Acquisition of data, Drafting or revising the article; MA, Final approval of the version to be published, Drafting or revising the article, Contributed unpublished essential data or reagents; YF, EM, SD, OAA, GAS, DAB, DK, RBB, Final approval of the version to be published, Conception and design, Drafting or revising the article; AMD, Final approval of the version to be published, Conception and design, Analysis and interpretation of data, Drafting or revising the article; AD, Final approval of the version to be published, Conception and design, Acquisition of data, Analysis and interpretation of data, Drafting or revising the article

## Author ORCIDs

Srdjan Djurovic, http://orcid.org/0000-0002-8140-8061
Gaute T Einevoll, http://orcid.org/0000-0002-5425-5012
Anna Devor, http://orcid.org/0000-0002-5143-3960

## Ethics

Animal experimentation: This study was performed in strict accordance with the recommendations in the Guide for the Care and Use of Laboratory Animals of the National Institutes of Health. All of the animals were handled according to approved institutional animal care and use committee (IACUC) protocols (#S07360, S14275) of the University of California San Diego.

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
