## [Decision Letter]

Thank you for submitting your article "Cell type specificity of neurovascular coupling in cerebral cortex" for consideration by *eLife*. Your article has been reviewed by two peer reviewers, including Bruno Cauli, and the evaluation has been overseen by Sacha Nelson as the Reviewing Editor and David Van Essen as the Senior Editor.

The reviewers have discussed the reviews with one another and the Reviewing Editor has drafted this decision to help you prepare a revised submission.

Summary:

Uhlirova and colleagues use optogenetic stimulation, 2-photon imaging and pharmacology to examine the contribution of different neuronal populations to neurovascular coupling in the mouse somatosensory cortex. The dissociation of dilation and constriction and the implication of NPY in this process are the strongest elements of the paper. Overall this is a very elegant and technically impressive study. While most studies in the field focused almost exclusively on the role of astrocytes, this group and others have recently challenged the contribution of this cell type in this process, and now provide compelling in vivo evidence for the importance of cortical neurons in neurovascular coupling. The finding that some subpopulations of cortical neurons are specialized for the control of the intracortical vasculature will have an important impact on the reliable interpretation on fMRI signals widely used to infer neuronal activity in health and disease.

Essential revisions:

1) There are inadequately supported claims, the most important of which involve not having recorded from neural firing directly. For example, the first section, with its indirect ruminations about what happens when the VGAT line is driven, has many points where leaps in inference (from light density to likely firing patterns to absence of dendritic effects) make the case less compelling. The reviewers both thought that the paper would be strengthened by including unit recordings (perhaps in a preparation without simultaneous vascular recording), however both reviewers and the editors agreed that doing this well (e.g. identifying the relevant interneurons) would be challenging and that therefore this was not absolutely required for acceptance. In the event that recordings are not included care should be taken to revise the text to acknowledge this missing line of evidence.

2) The inferences that the time course of the arteriolar response dictates the focus of the response initiation-that because the first onset is deep it must only initiate there-is tenuous. It could of course be a multi-tiered response, in accordance with the fact that the authors likely drove interneurons throughout the depth. The authors tiptoe around this and need to just state clearly that there could (and, likely are) many lamina-specific drivers.

3) While the authors mention the Anenberg, Murphy (2015) study, they should dive deeper into the differences (for example, that study did not report flow decreases – is this difference a volume vs. diameter vs. flow calculation difference across speckle vs. 2-photon, or is it about the time course of measurement, or is it a real disagreement about findings?) Any of these interpretations is fine from the perspective of the contribution of the current data, but I do want to know their take on this in print. Although a decrease in blood flow induced by OG stimulation of VGAT-ChR2 mice has not been reported per se in the Anenberg et al. 2015 study, looking closely at Figure 2 of this paper shows a positive fractional change of the speckle contrast (i.e. blood flow decrease) in the periphery of the activated region. This is particularly evident at 4s after the onset of stimulation. Given that the 1) "Regions of interest selected for assessment of speckle contrast over time were 1mm^[2]^ and centered at the site where the laser was targeted on the cortex" and 2) that most of the positive fractional change of the speckle contrast does not occur in these ROI, this might explain why Anenberg and colleagues have not reported a decrease in blood flow. This should be indeed acknowledged.

[Editors' note: further revisions were requested prior to acceptance, as described below.]

Thank you for submitting your article "Cell type specificity of neurovascular coupling in cerebral cortex" for consideration by *eLife*. Your article has been reviewed by Bruno Cauli and the evaluation has been overseen by a Reviewing Editor and David Van Essen as the Senior Editor.

As you will see below, there remain some issues that will require your attention before the manuscript can be accepted.

*Reviewer #1:*

In this revised version of the manuscript the authors have addressed most of my concerns. In particular, they made substantial efforts by providing new data including recordings of MUA, Ca^2+^ imaging and GAD67 immunohistochemistry of VGAT-ChR2 mice. Although these new data do not directly show that OG stimulation activates INs and silences PCs, they provide a body of evidence supporting it (e.g. evoked spiking activity during long OG stimulation with concomitant suppression of LFP, absence of Ca^2+^ transient in putative PCs during OG stimulation, expression of ChR2-YFP restricted to GAD67 positive cells). The new Figure 3 would benefit from insets/panels showing recordings of MUA during ongoing neuronal activity and during OG stimulation. Providing these recordings at a higher magnification and a better temporal resolution would allow better appreciation of spike shapes.

They have also better discussed the discrepancy with the Anenberg et al. 2015 (e.g. differences in the anesthetics used and stimulation paradigms). Surprisingly, and in spite of my suggestions, the authors do not consider at all that Anenberg et al. have not reported blood flow decrease induced by OG stimulation in VGAT-ChR2 mice simply because their analysis procedure could barely catch it.

---

## [Author Response]

*Essential revisions:*

1) There are inadequately supported claims, the most important of which involve not having recorded from neural firing directly. For example, the first section, with its indirect ruminations about what happens when the VGAT line is driven, has many points where leaps in inference (from light density to likely firing patterns to absence of dendritic effects) make the case less compelling. The reviewers both thought that the paper would be strengthened by including unit recordings (perhaps in a preparation without simultaneous vascular recording), however both reviewers and the editors agreed that doing this well (e.g. identifying the relevant interneurons) would be challenging and that therefore this was not absolutely required for acceptance. In the event that recordings are not included care should be taken to revise the text to acknowledge this missing line of evidence.

We agree that our manuscript was lacking direct demonstration that OG stimulation in VGAT mice causes firing of INs and not excitatory neurons. We also greatly appreciate the reviewers’ understanding that providing such evidence in vivo would be a challenging task. We made all efforts to strengthen our line of evidence in the following 3 ways:

A) Since SI cortex was not specifically examined in the original paper describing generation of the VGAT line, we confirmed expression of the EYFP-ChR2 construct in GABAergic INs. This new result is now included:

“Double immunofluorescence of ChR2-EYFP and GABA-producing glutamate decarboxylase in this mouse line revealed a virtually complete overlap of staining in the SI confirming that the ChR2-EYFP protein was indeed present only in GABAergic INs (Figure 2—figure supplement 1).”

B) We have included Multiple Unit Activity (MUA) data corresponding to the Local Field Potential (LFP) recordings shown in Figure 3. OG stimulation for the duration of 200 ms resulted in a sharp and transient increase in MUA timed to the stimulus onset (Figure 3). During prolonged OG stimulation for 1s, the initial transient response was followed by a desynchronized activity, starting ~200 ms after the stimulus onset and lasting for the duration of the stimulus (Figure 3). Since LFP during OG stimulus was largely suppressed, we speculate that this MUA response reflects firing of INs: initially, the OG stimulus depolarizes INs causing a synchronized MUA response. Then, synaptically connected INs inhibit each other; this is why the initial MUA response is transient. During persistent OG stimulation, synaptic connectivity between different types of INs in the absence of a closed circuit loop (no participation of excitatory neurons) eventually results in desynchronized IN firing reflecting a steady state balance between the ongoing OG stimulation and mutual inhibition. These new results are now described as follows:

“The MUA signal reflects spiking of all neurons, excitatory and inhibitory, within ~100 μm of the electrode tip (Buzsaki, 2004; Einevoll et al., 2013; Pettersen and Einevoll, 2008). […] This MUA response in the presence of LFP suppression suggests firing of INs rather than PCs.”

C) We performed in vivo 2-photon calcium imaging and combined calcium imaging with MUA recordings. Results of these experiments are illustrated in Figure 3—figure supplement 1 and described as follows:

“To provide further evidence that OG stimulation did not engage PCs, we performed in vivo calcium imaging in an additional 3 VGAT-ChR2(H134R)-EYFP subjects (Figure 3—figure supplement 1). […] This is because no calcium increase was detected in PCs in response to the OG stimulus, although spontaneous calcium transients were readily detectable.”

We also included a corresponding new section in the Methods:

“Identification of INs and PCs in vivo and combination of 2-photon calcium imaging and OG stimulation:

First, we acquired 2-photon Z-stacks of structural reference images. […] In this regime, we were in a good position to detect calcium transients that have a fast rise and slow decay (time constant ~1s (Lutcke et al., 2013)).”

2) The inferences that the time course of the arteriolar response dictates the focus of the response initiation-that because the first onset is deep it must only initiate there-is tenuous. It could of course be a multi-tiered response, in accordance with the fact that the authors likely drove interneurons throughout the depth. The authors tiptoe around this and need to just state clearly that there could (and, likely are) many lamina-specific drivers.

Yes, we agree that lamina-specific drivers are likely to be present and apologize for not clarifying this in the text. We have revised the relevant discussion as follows:

“It is well established that dilation and constriction can propagate along arteriolar walls (Liu et al., 2012; Perrenoud et al., 2012; Segal and Duling, 1986; Rosenblum, Weinbrecht and Nelson, 1990; Jensen and Holstein-Rathlou, 2013; Chen et al., 2014), and these conducted responses may have contributed to the spread of OG dilation and constriction throughout the cortical depth in our data. […] Therefore, substantial delays in the superficial arteriolar branches suggest that local neurovascular communication in the upper layers, if it exists, has slower kinetics.”

3) While the authors mention the Anenberg, Murphy (2015) study, they should dive deeper into the differences (for example, that study did not report flow decreases – is this difference a volume vs. diameter vs. flow calculation difference across speckle vs. 2-photon, or is it about the time course of measurement, or is it a real disagreement about findings?) Any of these interpretations is fine from the perspective of the contribution of the current data, but I do want to know their take on this in print. Although a decrease in blood flow induced by OG stimulation of VGAT-ChR2 mice has not been reported per se in the Anenberg et al. 2015 study, looking closely at Figure 2 of this paper shows a positive fractional change of the speckle contrast (i.e. blood flow decrease) in the periphery of the activated region. This is particularly evident at 4s after the onset of stimulation. Given that the 1) "Regions of interest selected for assessment of speckle contrast over time were 1mm^2^ and centered at the site where the laser was targeted on the cortex" and 2) that most of the positive fractional change of the speckle contrast does not occur in these ROI, this might explain why Anenberg and colleagues have not reported a decrease in blood flow. This should be indeed acknowledged.

We think that this disagreement of findings may be due to differences in anesthesia and/or stimulation protocol. The Murphy group used ketamine/xylazine, while we used α-chloralose. We feel confident about the constriction phase in our anesthetized measurements because our awake measurements replicate the same biphasic response behavior. The Murphy’s group implemented OG stimulation at 100Hz while in our case light was applied in ~200-ms long pulses. We have added the following paragraph to the Discussion:

“OG stimulation in the VGAT-ChR2(H134R)-EYFP line produced a monophasic CBF increase in a recent laser speckle contrast imaging study by Anenberg et al. (Anenberg et al., 2015). […] In contrast to other types of INs driving dilation, at least some NPY-positive neurons exhibit a pronounced spike latency (Karagiannis et al., 2009) and thus may not respond well to short (5-ms duration) light pulses delivered at high frequency (100 Hz).”

We also added the following sentence where we discuss limitations of anesthesia:

“As such, the ketamine/xylazine anesthesia employed by Anenberg et al. (Anenberg et al., 2015) may provide an additional explanation for the discrepancy in our results regarding the constriction phase in response to OG stimulation of INs.”

Regarding their Figure 2 decrease in the periphery is present in the images taken at 2 and 4s after the stimulus onset but not in the middle image taken at 3s after the stimulus onset. This intermittent appearance of the negative signal suggests to us that it may be dominated by noise rather than representing a real physiological behavior.

[Editors' note: further revisions were requested prior to acceptance, as described below.]

*As you will see below, there remain some issues that will require your attention before the manuscript can be accepted.*

*Reviewer #1:*

In this revised version of the manuscript the authors have addressed most of my concerns. In particular, they made substantial efforts by providing new data including recordings of MUA, Ca^2+^ imaging and GAD67 immunohistochemistry of VGAT-ChR2 mice. Although these new data do not directly show that OG stimulation activates INs and silences PCs, they provide a body of evidence supporting it (e.g. evoked spiking activity during long OG stimulation with concomitant suppression of LFP, absence of Ca^2+^ transient in putative PCs during OG stimulation, expression of ChR2-YFP restricted to GAD67 positive cells). The new Figure 3 would benefit from insets/panels showing recordings of MUA during ongoing neuronal activity and during OG stimulation. Providing these recordings at a higher magnification and a better temporal resolution would allow better appreciation of spike shapes.

The reviewer’s thought is that fast spiking inhibitory neurons produce extracellular signals that are narrower compared to PCs (Bartho, Buzsaki, J Neurophysiol 92: 600-608, 2004). Therefore, if we could resolve individual spikes during the OG stimulus, the expectation is that their average waveform would be narrower compared to that during spontaneous activity where PCs affect the signal duration. However, identification/isolation of individual spikes and spike sorting with extracellular MUA recordings in vivo is a difficult problem. Usually, it is addressed using electrode arrays (e.g., so called “tetrodes”) such that a spike from a given cell can be picked up by a number of recording contacts increasing the fidelity of isolation procedure. In our experiments, we used a single-contact electrode that has sensitivity to spikes within a ~100 μm sphere around the electrode tip, containing many neurons of different types with their relative signal amplitudes depending on the size of the soma and distance from the electrode tip. In addition, cerebral circuits in vivo (at least, under anesthesia), experience up and down states with bursts of spiking activity during the up state resulting in overlapping spike waveforms. Under these conditions, isolation of individual spikes is an inherently ambiguous problem.

Following the reviewer’s comment, we have attempted isolation of spikes during the spontaneous activity and during OG stimulation. After visual inspection of the data, we defined a range of event amplitudes that may correspond to single spikes and aligned these events by their troughs. Comparison of these putative single-spike events from the spontaneous and OG category shows no significant difference in the width of the wavelet, see Figure 7. However, we believe that this result cannot be interpreted in a meaningful way due to the following considerations, leading to smearing of differences:

a) We have no confidence that the events we chose for the analysis indeed correspond to single spikes. Some of them may be a number of spikes slightly shifted in time. Depending on the amount of shift, this can lead to either broadening or narrowing of the waveform;

b) Not all inhibitory neurons are fast spiking, leading to further smearing of potential differences when considered as a population;

c) Both PCs and INs participate in spontaneous circuit activity, therefore our sample of putative spontaneous spike events has a mixed origin.

Given these constraints and ambiguity, we believe that the results of our analysis are inconclusive and would not be helpful for the reader. Addressing the issue of spike identification and sorting comprehensively would require switching to a different type of electrodes (tetrodes) plus solving the problem of the insertion of these (thicker) electrodes under the cover glass into the brain tissue without damaging the optical conditions. This type of effort would require a significant investment and probably is outside the current scope. Miniaturization of electrode arrays is one of the ongoing efforts for the BRAIN Initiative here in the US, and is likely to largely facilitate this kind of studies in the near future.

Author response image 1.**DOI:**
http://dx.doi.org/10.7554/eLife.14315.016

They have also better discussed the discrepancy with the Anenberg et al. 2015 (e.g. differences in the anesthetics used and stimulation paradigms). Surprisingly, and in spite of my suggestions, the authors do not consider at all that Anenberg et al. have not reported blood flow decrease induced by OG stimulation in VGAT-ChR2 mice simply because their analysis procedure could barely catch it.

Assuming that we are continuing the discussion about the peripheral signal in Anenberg et al. Figure 2, we remain of the opinion that this signal is likely to be an artifact, possibly resulting from varying baseline correction leading to changing bias from frame to frame. In addition, the sequence of images in Figure 2 is inconsistent with the time-course shown in Figure 2. Specifically, the time-course in Figure 2 shows a monotonic decrease in CBF between 3 and 7s post-stimulus. However, careful inspection of the central part of the images (used for extraction of time-courses) suggests that CBF at 6s is higher than that at 5s. Furthermore, the entire image at 6s is yellow color-shifted (i.e., an increase in CBF) including the periphery. These spatially coherent fluctuations of the peripheral intensity from frame to frame suggest to us that this is not due to the measurement noise but incorrect procedure of image analysis. Since Anenberg et al. raw data are not available to us, and their image analysis procedure is not described in sufficient details, we can only speculate about what exactly went wrong with their analysis. Therefore, we prefer not to include specific comments about this peripheral signal in our manuscript.